# Theoretical Study of Electronic and Thermal Transport Properties through a Single-Molecule Junction of Catechol

Erika Y. Soto-Gómez [1,2,*], Judith Helena Ojeda Silva [1,3], John A. Gil-Corrales [4], Daniel Gallego [1], Mikel F. Hurtado Morales [5], Alvaro L. Morales [4] and Carlos A. Duque [4]

1   Grupo de Investigación Química-Física Molecular y Modelamiento Computacional (QUIMOL), Facultad de Ciencias, Universidad Pedagógica y Tecnológica de Colombia, Tunja 150003, Colombia; judith.ojeda@uptc.edu.co (J.H.O.S.); daniel.gallego@uptc.edu.co (D.G.)

2   Grupo de Investigación en Ciencias Básicas Aplicación e Innovación (CIBAIN), Facultad de Ciencias, Universidad Internacional del Trópico Americano (Unitrópico), Yopal 850001, Colombia

3   Grupo de Física de Materiales, Facultad de Ciencias, Universidad Pedagógica y Tecnológica de Colombia, Tunja 150003, Colombia

4   Grupo de Materia Condensada-UdeA, Instituto de Física, Facultad de Ciencias Exactas y Naturales, Universidad de Antioquia UdeA, Calle 70 No. 52-21, Medellín 50010, Colombia; jalexander.gil@udea.edu.co (J.A.G.-C.); alvaro.morales@udea.edu.co (A.L.M.); carlos.duque1@udea.edu.co (C.A.D.)

5   Facultad de Ingeniería, Departamento de Ingeniería Civil, Corporación Universitaria Minuto de Dios, Bogotá 111021, Colombia; mikel.hurtado@uniminuto.edu

*   Correspondence: erikayazmin.soto@uptc.edu.co

**Abstract:** The study of molecular nanoelectronic devices has recently gained significant interest, especially their potential use as functional junctions of molecular wires. Aromatic systems with $\pi$-conjugated bonds within their chemical backbones, such as catechol, have attracted particular attention in this area. In this work, we focused on calculating and determining catechol's electrical and thermal transport properties using the theoretical method of Green's functions renormalized in a real space domain within a framework of tight-binding approximation to the first neighbors. Thus, we studied two theoretical models of catechol as a function of its geometry, obtaining striking variations in the profiles of electrical and thermal conductance, the Seebeck coefficient, and the figure of merit. The analyses of the results suggest the potential application of catechol as a likely conductive and thermoelectric molecule serving as a novel material to use in molecular electronic devices.

**Keywords:** molecular electronic; catechol; transmission probability; thermoelectrics





## 1. Introduction

During the last decades of the new millennium, electronic device miniaturization reached the frontier in the nanoscale domain, an area most commonly known as nanoelectronics. At this scale, researchers demonstrated that size matters in increasing efficiency and high-throughput information processing, leading to large-scale consumption. Due to its high abundance and great performance in such devices, silicon is used as a semiconductor material; however, the high level of purity required for its application and overheating make its industrial application non-sustainable due to its charge losses, toxic environmental trail, and costs. For this reason, nanotechnology researchers are strengthening efforts to generate novel materials as substituents of silicon in this area [1,2] and apply them in new electronic devices, such as switches, diodes, rectifiers, solar cells, field effect transistors, and quantum wires, among other applications [3–5], as well as in molecular electronics.

In searching for such devices, experimental and theoretical approaches have been reported in the literature over the years. The seminal work of Mann and Kuhn in 1971 [6] demonstrated the electrical properties of cadmium carboxylate salt derivatives of fatty

acids through the measurement of conductivity in a monolayer array of this system. In addition, Cooper et al. in 1971 found conductive behavior at low temperatures using a molecular system of tetrathiofuvalene ($TTF$) [7,8], and in the late 1970s, Macdiarmid et al. reported the conductive behavior of organic polymers [9,10], an effort recognized by the Nobel Prize in Chemistry in 2000.

On the other hand, instrumental technical advances in the 1980s led to robust analytical techniques, such as scanning tunneling microscopy (STM) and atomic force microscopy (AFM) [11], allowing measurements of electrical properties of several nanoscopic systems, including molecules. However, some scenarios presented difficulties at the junctions of molecules with the tip probe due to the intrinsic conditions of the techniques for such malleable and soft systems. Fortunately, these issues were solved with the development of the mechanically controllable break junction (MCBJ) [12], the scanning tunneling microscopic break junction (STMBJ), the electromigrated break junction (EBJ), and the thermoelectric atomic force microscope (ThAFM), thus enabling more precise data of electrical and thermal conductance in a single-molecule junction, leading to the invention of nanoscale organic thermoelectric devices [2,13]. Thus, in 2011, the conjugated polymer poly(3,4-ethylenedioxythiophene) (PEDOT) was characterized as one of the first systems with moderate thermal-to-electrical energy conversion efficiency [14]. Since then, researchers have focused their efforts on increasing this conversion efficiency through the fabrication of new organic thermoelectric materials [15]. 3,6-Disubstituted catechol has been used as the base of highly functionalized molecular rods possessing interesting electrical transport properties [16]. In another study, poly-substituted catechol was used as an end arm of a long-carbon-chain hemiquinones (i.e., catechol covalently linked with ortho-benzoquinone), and STM analysis showed that catechol acts mainly as the donor, whereas the ortho-quinone serves as the charge acceptor [17].

Likewise, circular wires were built by a catechol bridge between highly conjugated skeletons composed of aromatic rings and alkyne moieties covalently linked to two terminals. The redox pair catechol/quinone served as a molecular switch in an STM assembly, and thus, the researchers demonstrated that the electronic transport properties strongly depend on the oxidation state, which can be tuned by the degree of electronic delocalization and, therefore, the electronic conductance of the systems [18]. In this work, Nicolas Weibel et al. set up the design and fabrication of potential advanced active polymeric materials, such as catechol-based organic electrodes containing reversible redox sites, which could be applied for economical and sustainable next-generation electrochemical energy storage (EES) devices [19]. In addition, the 1,2-dihydroxybenzene scaffold is essential in a wide variety of organic dyes, typically large organic molecules, for this optical behavior. Therefore, despite the experimental evidence on how catechol plays a crucial role in the intramolecular electronic properties of such large molecules, there is a scarce fundamental understanding of its internal electronic structural features and how electronic transport proceeds within such systems [20–22].

In terms of theoretical studies, the seminal work conducted by Aviram and Ratner in 1974 on electrical properties through organic molecules [23] demonstrated their potential to act as possible rectifiers by studying the electrical and thermal transport processes in individual molecules, evidencing the essential role of aromatic systems in the semiconducting properties of these molecules [24,25]; however, such properties may vary depending on the configuration in which they are held between the metal contact or tips (i.e., symmetric or asymmetric). In addition, the electrical and magnetic properties can vary drastically [26,27], taking to account (or not) external stimuli such as impurities, vacancies, electric or magnetic fields, and temperature variations, among others.

Thus, we sought to study the electrical and thermal transport properties of catechol to understand how these transport processes depend on several variables and to determine its potential application in molecular conducting devices. To this end, we modeled catechol in the middle of two electrodes in two configurations (Model I and Model II) and calculated the electrical and thermal transport properties employing the Green's function method

through the use of the Dyson dynamic equation, based on a tight-binding Hamiltonian, within the renormalization framework or the decimation of the real system to an effective system. Then, we focused our interest on the variation in the position of the electrodes and the electrode–molecule coupling bond (strong and weak regimes) to calculate the properties of the transmission probability, the electric current, the electric conductance, the thermal conductance, the Seebeck coefficient, and the ZT factor. Very interestingly, we found significant differences between them, as described herein. The outline of this work is as follows. In Section 2, we discuss the details of the analytical model used to determine the transport properties of the molecular system. In Section 3, we describe the method, followed by Section 4, where we present and describe the results of the transport properties. Finally, in Section 5, we highlight the main conclusions of the work and provide final remarks.

## 2. Model

1,2-Dihydroxybenzene is an organic molecule commonly known as catechol with the molecular formula $C_6H_6O_2$ (Figure 1). Due to its molecular and electronic structures, researchers have focused their interest on using it as photosensitizer in photovoltaic devices supported over anatase particles as a semiconducting substrate, which is normally used as such in this research area [28].

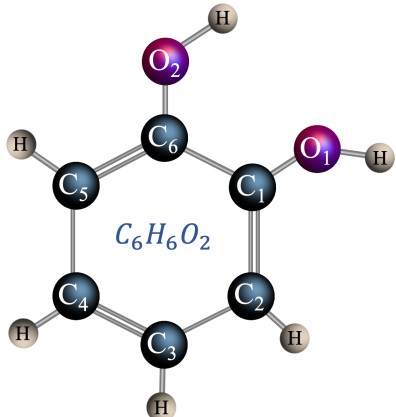

**Figure 1.** Molecular structure of catechol.

Based on catechol's versatility and physicochemical properties, particularly its thermo-electric behavior, we sought to use it in an electronic device as a conducting molecular wire. Thus, we modeled catechol's structure (C: carbon atoms; OH: hydroxyl groups) between two electrodes (L: left; R: right), where its linked sites have their respective electrochemical potentials $\mu_L$ and $\mu_R$ in Models I and II (Figures 2a and 3a).

When an electrochemical potential difference is applied between the two electrodes ($\mu_R - \mu_L = -eV$), one electrode turns into a source of electrons, and its counterpart acts as an electron drain. This dynamic process occurs because, under an external stimulus, the system is out of equilibrium, causing a current to flow through an electric circuit that closes the system [29]. Thus, we describe the models shown in Figures 2a and 3a by the nearest-neighbor tight-binding Hamiltonian [30–32]:

$$H = H_{cat} + H_L + H_I, \tag{1}$$

where $H_{cat}$ is the Hamiltonian of the catechol molecular system, described by

$$H_{cat} = \sum_i t_i(c_i^\dagger c_{(i+1)} + c_{(i+1)}^\dagger c_i) + \sum_i E_i c_i^\dagger c_i, \tag{2}$$

where $t_i$ is the coupling between the atomic sites of the system, represented in this model as $\Omega$ for a single bond ($\sigma$-bond) between C-C atoms, $\Pi$ for a double bond ($\pi$-bond) between

C=C atoms, and $\Lambda$ for a $\sigma$-bond between carbon and a hydroxyl group (C-OH). $E_i$ represents the energy of the atomic sites, denoted in this case as $E_C$ and $E_{OH}$ for the C and OH atomic sites, respectively. And $c_i$ ($c_i^\dagger$) is the creation (destruction) operator of the electron at atomic site $i$.

Assume that $H_L$ is the Hamiltonian of the leads and $H_I$ is the Hamiltonian of interaction between the leads and the molecule, which are given by

$$H_L = \sum_{k_L} \varepsilon_{k_L} d_{k_L}^\dagger d_{k_L} + \sum_{k_R} \varepsilon_{k_R} d_{k_R}^\dagger d_{k_R} ,\tag{3}$$

and

$$H_I = \sum_{k_L} \Gamma_L d_{k_L}^\dagger c_1 + \sum_{k_R} \Gamma_R d_{k_R}^\dagger c_N + h.c.\tag{4}$$

where the creation (destruction) operators of an electron in state $k_{L,R}$ are given by $d_{k_{L,R}}$ ($d_{k_{L,R}}^\dagger$), with energy $\varepsilon_{k_{L,R}}$. $\Gamma_{L,R}$ denotes the coupling between each electrode and the molecular system, and $h.c.$ is the Hamiltonian's complex conjugate.

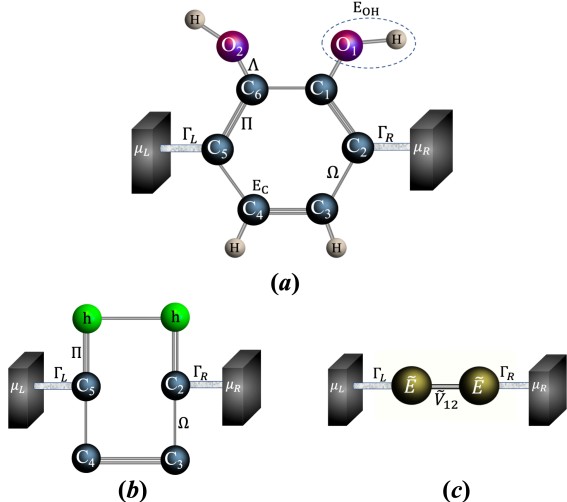

**Figure 2.** Model I. (**a**) Catechol molecular system, (**b**) reduced geometric model, and (**c**) effective linear chain.

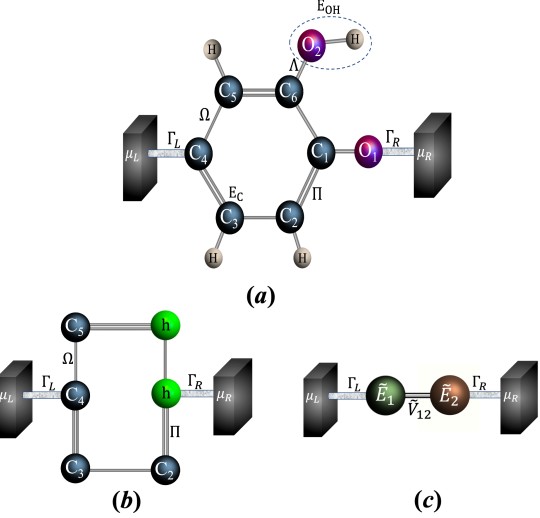

**Figure 3.** Model II. (**a**) Catechol molecular system, (**b**) reduced geometric model, and (**c**) effective linear chain.

### 3. Method

The transmission probability ($T(\varepsilon)$) is calculated using the Green's function of the molecular system, where the catechol's electrical and thermal transport properties are thus determined. This Green's function is derived by applying the Dyson equation within a real-space renormalization or decimation scheme [25,29,33,34].

#### 3.1. Transmission Probability

The transmission probability ($T(\varepsilon)$) is defined through the Fisher–Lee relationship, expressed as:

$$T(\varepsilon) = \text{Tr}[\Gamma_L G^r \Gamma_R G^a], \tag{5}$$

where $\text{Tr}(A)$ denotes the trace of the matrix $A$, which depends of the retarded ($G^r$) and advanced ($G^a$) Green's functions of the system. The electrode couplings $\Gamma_{L(R)}$ are functions of the spectral density matrix of the left (right) contacts, respectively, given by $\Gamma_{L(R)} = i[\Sigma_{L(R)} + \Sigma_{L(R)}]$.

However, in this work, we calculate $T(\varepsilon)$ for effective linear chains (Figures 2c and 3c); thus, Equation (5) turns into the following:

$$T(\varepsilon) = \Gamma_{11}^L \Gamma_{NN}^R |G_{1N}|^2, \tag{6}$$

where $G_{1N}$ is determined by:

$$G_{1N} = \frac{G_{1N}^0}{(1 - G_{11}^0 \Sigma_L)(1 - G_{NN}^0 \Sigma_R) - G_{N1}^0 G_{1N}^0 \Sigma_R \Sigma_L}. \tag{7}$$

Now, to calculate the Green's functions $G_{1N}^0$, $G_{11}^0$, $G_{N1}^0$, and $G_{NN}^0$, it is necessary to apply the renormalization process to each model; however, it is worth recalling that for the models analyzed here, $G_{1N}^0 = G_{N1}^0$ and $G_{11}^0 = G_{NN}^0$.

#### 3.2. Renormalization Process

Green's functions can be defined as the solutions of inhomogeneous differential equations of the type:

$$[z - H(r)]G(r, r', z) = \delta(r - r'), \tag{8}$$

where $r$ and $r'$ are subject to certain boundary conditions, with complex $z$ given by $z = \varepsilon + i\eta$, which depends on the energy of the electron ($\varepsilon$) when it enters the system, and $\eta$ is an infinitesimal term. $H(r)$ is the Hamiltonian operator (hermitian), which is time-independent, having a complete set of eigenfunctions $\phi_n(r)$ that satisfy the same boundary conditions as $G(r, r', z)$ and can be considered orthonormal without loss of generality.

From Equation (5), the obtained Green's functions are expressed in terms of H as $G = 1/(z - H)$, and taking into account some algebraic processes, a dynamic equation known as the Dyson equation can be obtained. Thus, we take the latter as a starting point for this work since it has previously been used for some one-dimensional and quasi-dimensional models. The Dyson equation is defined as:

$$G = G_0 + G_0(\Sigma_L + \Sigma_R)G, \tag{9}$$

where $G_0$ is the Green's function of an isolated system, and $\Sigma_{L(R)}$ is the self-energy of the left (right) electrode. Thus, we make use of the Dyson equation in order to reduce the catechol molecular system (Model I—Figure 2a), which has two degrees of freedom due to its planar nature, to an effective linear system with one degree of freedom (Model I—Figure 2c). However, to simplify the procedure, a geometrical scheme is formulated (Figure 2b) with the proper label in each atomic site, thus having the respective numbering in each Green's function. After renormalizing each model to an effective linear chain, we

obtain linear chains with two atomic sites with effective energies $\tilde{E}$ and effective local Green's functions $\tilde{g}$ (see Appendix A). For Model I, $\tilde{g}$ is given by:

$$\tilde{g} = \frac{g_c(-1+g_h^2\Omega^2)(-1+g_c^2\Pi^2)}{-g_h^2\Omega^2 + \alpha + g_h^3 g_h \Pi^4 + g_c^2(-1+g_h^2\Omega^2)(\Omega^2+\Pi^2)} ,\tag{10}$$

where $\alpha = 1 - g_c g_h \Pi^2$. In addition, the effective coupling between these effective sites is also calculated and given by:

$$\tilde{V}_{12} = \frac{\Pi\Omega[g_h^2\Pi + g_c^2(\Omega - g_h^2\Omega^3 - g_h\Pi^3)]}{(-1+g_h^2\Omega^2)(-1+g_c^2\Pi^2)} ,\tag{11}$$

As regards Model II, we obtain an effective linear chain with two effective atomic sites with two distinct effective energies, $\tilde{E}_1$ and $\tilde{E}_2$, and two particular effective local Green's functions, $\tilde{g}_1$ and $\tilde{g}_2$, respectively (see Appendix A), given by:

$$\tilde{g}_1 = \frac{g_c(-1+g_c g_h \Pi^2)(-1+g_c^2\Omega^2)}{\alpha + g_c^4\Omega^4 + g_c^3 g_h \Pi^2(\Pi^2+\Omega^2) - g_c^2(\Pi^2+2\Omega^2)} ,\tag{12}$$

and

$$\tilde{g}_2 = \frac{g_h(-1+g_c g_h \Pi^2)(-1+g_c^2\Omega^2)}{1 - 2g_c g_h \Pi^2 - g_h^2\Omega^2 + g_c^3 g_h \Pi^2\Omega^2 + g_c^2(-\Omega^2 + g_h^2\gamma)} ,\tag{13}$$

where $\gamma = (\Pi^4 + \Omega^4)$. The effective coupling for Model II is calculated and given by:

$$\tilde{V}_{12} = \frac{g_c\Pi\Omega(g_c\Pi + g_h\Omega - g_c^2 g_h(\Pi^3+\Omega^3))}{(-1+g_c g_h \Pi^2)(-1+g_c^2\Omega^2)} ,\tag{14}$$

Once we calculate and determine the effective local Green's functions with their corresponding effective couplings for each model, we proceed to determine the Green's functions $G_{1N}$ and $G_{NN}$, given by Equation (6) (see details in Appendix A), and calculate their respective transmission probabilities $T(\varepsilon)$.

*3.3. Electrical and Thermal Transport Properties*

To determine the quantum transport properties or the flow of charge carriers through a system connected to two electrodes, we make use of the Landauer–Büttiker formalism, where the electric current is calculated by the expression:

$$I = \frac{2e}{\hbar} \int_{-\infty}^{\infty} (f_L - f_R)T(\varepsilon)d\varepsilon ,\tag{15}$$

where $e$ is the charge of the electron, $\hbar$ is the reduced Plank's constant, and $f_{L(R)}$ is the left (right) Fermi function, given by:

$$f_{L(R)}(\varepsilon) = \frac{1}{1 + exp\left(\frac{(\varepsilon - \mu_{L(R)})}{k_B\Theta}\right)} ,\tag{16}$$

where $\mu_{L(R)} = \varepsilon_F \pm eV/2$, in which $\varepsilon_F$ represents the Fermi energy, $k_B$ is Boltzmann's constant, and $\Theta$ is the equilibrium temperature.

Accordingly, we use the Landauer integrals (Equation (17)) to evaluate the electrical conductance and thermal transport properties.

$$\zeta_n = -\int T(\varepsilon)(\varepsilon - \varepsilon_F)^n \left(\frac{\partial f(\varepsilon)}{\partial\varepsilon}\right)d\varepsilon ,\tag{17}$$

Thus, the electrical conductance ($\mathcal{G}$), thermal conductance ($\kappa$) (defined as the energy transported through the system in the form of heat [35,36]), Seebeck coefficient (S), and

efficiency of a thermoelectric material to convert thermal energy into electrical energy, known as the figure of merit or *ZT* factor, are determined as follows:

$$\mathcal{G} = \frac{2e^2}{h} \zeta_0 .$$

(18)

$$\kappa = -\frac{2}{h\Theta} \left( \zeta_2 - \frac{\zeta_1^2}{\zeta_0} \right),$$

(19)

$$S = -\frac{1}{e\Theta} \frac{\zeta_1}{\zeta_0},$$

(20)

$$ZT = \frac{\mathcal{G}S^2\Theta}{\kappa} = \frac{1}{\frac{\zeta_0\zeta_2}{\zeta_1^2} - 1},$$

(21)

Summing up, Equations (18)–(21) show the dependence of the electrical and thermal properties on the probability of transmission. Notably, the energy transport in the form of heat ($\kappa$) depends on the sum of the electrical ($\kappa_{el}$) and phononic ($\kappa_{ph}$) contributions. However, for the molecular systems studied in this work, $\kappa_{el}$ is much larger than $\kappa_{ph}$; therefore, the contributions of $\kappa_{ph}$ to thermo-electric transport are negligible [35,37].

### 4. Results

In order to understand the conducting behavior of Models I and II, we evaluated the transmission probability ($T(\varepsilon)$) as a function of the energy injected into the molecular system (Figure 4), in contrast to a simpler aromatic system without any functionality (i.e., benzene), for values of $\Gamma = 0.2$ eV, $E_C = 0.0$ eV, $E_{OH} = 0.5$ eV, $\Omega = 1.0$ eV, $\Pi = 1.0$ eV, and $\Lambda = 1.0$ eV. The resonant peaks occurring at weak coupling (i.e., $\Gamma < \Omega, \Lambda$) are closer to the eigenvalues of each molecular system, which are calculated by diagonalizing the Hamiltonian matrix $N \times N$ (Equation (3)), where $N$ corresponds to the number of atomic sites $N = 8$ (6 carbon atoms and 2 atoms of the OH groups). Then, with the eigenvalues given at 2.26 eV, $-2.15$ eV, 1.37 eV, 1.33 eV, $-1.24$ eV, $-1.13$ eV, 0.57 eV, and 0.0 eV (see Appendix A), we validated the renormalization for this molecular system [27,38] for the further determination of the electrical and thermal properties.

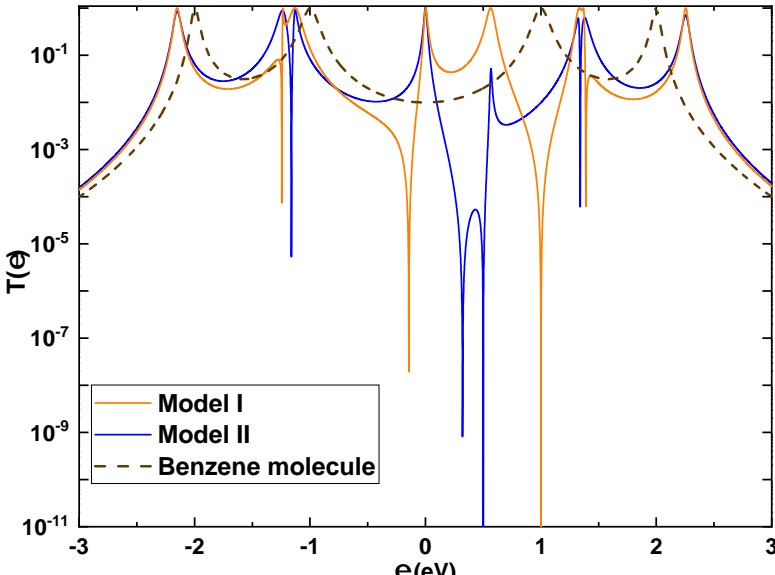

**Figure 4.** Transmission probability as a function of the energy for Models I (orange line) and II (blue line) from catechol and benzene (dashed brown line) molecular systems.

The resonant peak of $T(\varepsilon)$ around the Fermi level ($\varepsilon = 0$) for both models represents the allowed electronic states at that level, thus clearing the path for the transport of charge carriers due to an increase in the density of states. Therefore, the forbidden band between the highest occupied molecular orbital (HOMO) and the lowest unoccupied molecular orbital (LUMO) tends to cancel out, pointing out that, under these conditions, catechol acts as a conducting material. This result contrasts with the simpler aromatic system of the benzene molecule, which behaves as a semiconducting material (Figure 4, brown dashed line) with a gap around the Fermi level $\varepsilon_F = 0$ eV [24,25]. Since the OH groups alter the hybridization of the aromatic ring's orbitals of the molecular system, this clearly enhances the density of states around the Fermi level, turning it from a semiconducting to a conducting molecular skeleton.

When comparing the distinct behaviors of the two models, we can find evidence that the manner in which catechol's OH groups are anchored to the leads alters the resistance of the conducting molecular system. Thus, the resonant peaks in Model I are broader than those in Model II, although their eingenvalues are very close. This difference is strongly related to the fact that in Model I, none of the OH groups are attached to the leads, indicating a greater probability of the charge carriers diffusing along the molecular system, whereas, with one OH group attached to a lead (Model II), we can observe a lower area under the curve, indicating a more significant resistance to conductivity.

Because $T(\varepsilon)$ strongly depends on the coupling potential of the molecule to the leads ($\Gamma$), we screened it using different values, with frontier values of 0.0 eV to 3.0 eV (Figure 5), recalling that strong- and weak-coupling regimes are where $\Gamma > \Omega, \Lambda$ and $\Gamma < \Omega, \Lambda$, respectively. Figure 5 depicts the behavior of $T(\varepsilon)$ for different coupling values, highlighting selected cases of weak coupling ($\Gamma = 0.2$ eV, green line) and strong coupling ($\Gamma = 2.0$ eV, red line) regimes. Hence, the areas under the red line curves are broader than those under the green lines, showing that in the strong-coupling regime, the resonant peaks are broadened due to hybridization between the discrete states of the molecular system and the delocalized states of the contacts, inducing an increment in conductivity. In addition, Model I presents higher transmission than Model II due to the manner in which catechol is anchored to the leads, as previously discussed.

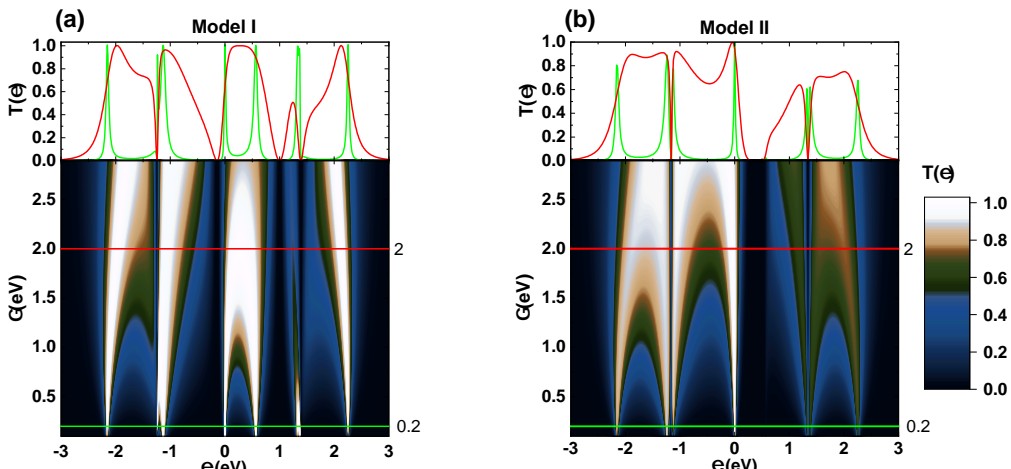

**Figure 5.** Transmission probability as a function of $\Gamma$ and energy for Models (**a**) I and (**b**) II of the catechol molecular system.

Once we defined the conducting behavior of the molecular device under weak- and strong-coupling regimes, we studied the current passing through it depending on the applied voltage between the leads (Figure 6). We observe lower currents with $\Gamma = 0.2$ eV for both models, in agreement with its semiconductor behavior, than with $\Gamma = 2.0$ eV when their conducting properties are enhanced. Notably, the current trend is similar in both models, with a larger amplitude for Model I (i.e., larger area under the curve),

with an interesting stepped shape owing to the resonance peaks previously shown for the transmission probability (Figure 4). The inset of Figure 6 shows the electric current within the voltage limits from −0.5 V to 0.5 V, showcasing the very similar behaviors of the two models at weak coupling but differences at strong coupling, as well as a more linear performance canceling out the stepped shape, which remains at weak coupling. This linear behavior in both models is characteristic of conductive materials.

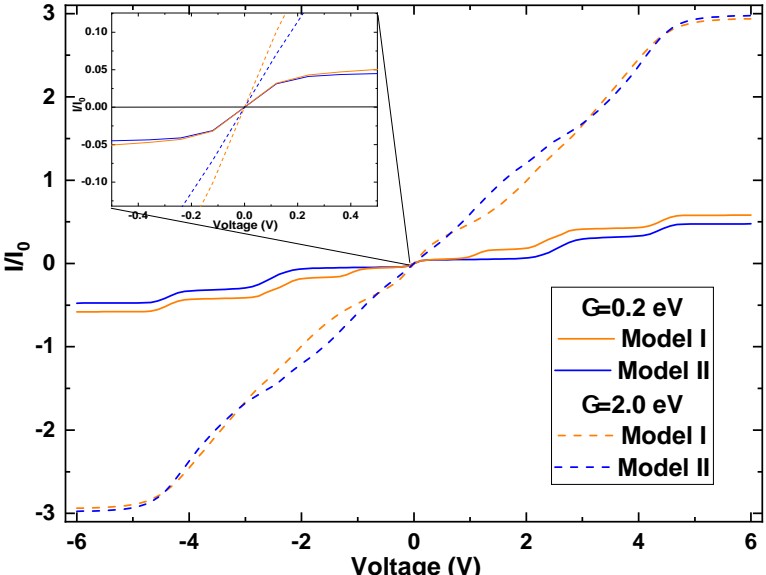

**Figure 6.** Current as a function of voltage for $\Gamma = 0.2$ eV (solid curves) and $\Gamma = 2.0$ eV (dashed curves) for both models (I and II), taking the temperature value $\Theta = 300$ K.

For the analysis of the thermal properties of conduction in both models, we determined the electrical conductance ($\mathcal{G}$), thermal conductance ($\kappa$), Seebeck coefficient ($S$), and figure of merit ($ZT$) as a function of the Fermi energy (Figure 7). Figure 7a shows that $\mathcal{G}$, calculated by Equation (18), agrees with the transmission (see above, Figure 5) according to Landauer's formalism, where the conductance is proportional to $T(\varepsilon)$ at low temperature and in the weak-coupling regime. Even though the conductance is pretty similar in both models, remarkably, at 0.5 V, Model II presents almost null conductance owing to the manner in which OH is anchored to the leads, suggesting the formation of an antiresonance state. Likewise, in Figure 7b, we observe a similar trend in $\kappa$ to that in $\mathcal{G}$ in both models, noticing slight differences in thermal conductance, indicating a similar heat transfer capacity in both models throughout the calculated energy range.

The performance of the thermopower or $S$ as a function of the Fermi energy is very intriguing (Figure 7c) since, around $\varepsilon_F = 0$ eV, a clear difference is observed for both models. To explain this behavior, let us look back at Figure 7a, where $\mathcal{G}$, and thus $T(\varepsilon)$ (Figure 4), becomes more asymmetric for Model II than for Model I around $\varepsilon_F = 0$ eV. This asymmetry causes the high value of $S$, creating a strong dependence on it concerning the asymmetric nature of the transmission or $\mathcal{G}$. Consequently, the higher the $S$ is, the larger the $ZT$ will be due to their proportionality (see Equation (21)). Therefore, to look for high thermoelectric efficiency (i.e., large $S$ value), we must seek high antisymmetrical transmission within the system. Thus, analyzing Figure 7c, we observe that for energies far from $\varepsilon_F = 0$ eV, the amplitude of $S$ decreases, and the transmission becomes more symmetrical or coherent [39].

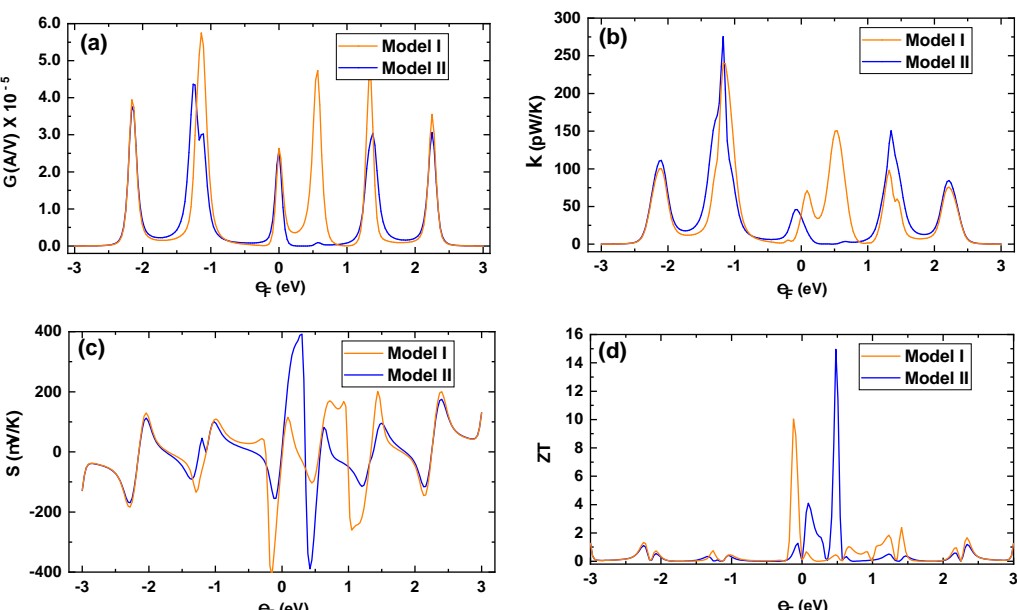

**Figure 7.** (**a**) Electrical conductance ($\mathcal{G}$), (**b**) thermal conductance ($\kappa$), (**c**) Seebeck coefficient ($S$), and (**d**) figure of merit ($ZT$) as a function of the Fermi energy ($\varepsilon_F$) for Models I (orange line) and II (blue line), taking values of temperature of $\Theta = 300$ K and coupling $\Gamma = 0.2$ eV.

Finally, in Figure 7d, we depict the $ZT$ factor with the same energy range as in the previous graphics in Figure 7. The fundamental differences between the two models occur for energies close to 0 eV. The maximum value reached for Model I is circa $ZT = 10$, while for Model II, the maximum value obtained is roughly $ZT = 15$, in accordance with the $S$ profile, where Model II has a higher maximum $S$ than Model I around these energies due to its asymmetry.

Thus, in Figure 7a, we can observe marked differences between Models I and II at the value of $\varepsilon_F = 0.5$ eV in the electrical conductance ($\mathcal{G}$), thermal conductance ($\kappa$), Seebeck coefficient ($S$), and figure of merit ($ZT$). This result strongly agrees with the transmission probability (Figure 4) since its value evaluated at $\varepsilon = 0.5$ eV presents antiresonance for Model II, where the transmission probability decreases considerably and reaches almost zero, whereas Model I has non-zero transmission. In this sense, according to Equations (18)–(21), the electrical and thermal properties depend on the transmission integral ($\zeta_n$) evaluated on the Fermi energy; therefore, at $\varepsilon_F = 0.5$ eV, the largest differences between the models are shown for each of these properties (Figure 7).

To evaluate the dependence of electrical conductance ($\mathcal{G}$) on the Fermi energy ($\varepsilon_F$) and the temperature ($\Theta$), we studied its variation in defined intervals of interest within a weak-coupling regime (Figure 8a,b). We observe that at low temperatures, the electrical conductance is higher owing to the lower resistance for the passage of electrons through the molecule due to less vibrational motion of the atoms within its chemical skeleton. Thus, at higher temperatures, the resistance is higher and the electrical conductance decreases. This pattern is evident for both models, depicting similar behavior. On the other hand, in Figure 8c,d, we present the maximum value of $\mathcal{G}$ as a function of temperature at $\Gamma = 0.2$ eV and as a function of $\Gamma$ at $\Theta = 300$ K, respectively. For this analysis, we have studied the variation in the Fermi energy within a window of ($-0.5$ eV $< \varepsilon_F < 0.5$ eV), because the most important behavior is observed when the Fermi energy is fixed around the inner edges of the allowed bands.

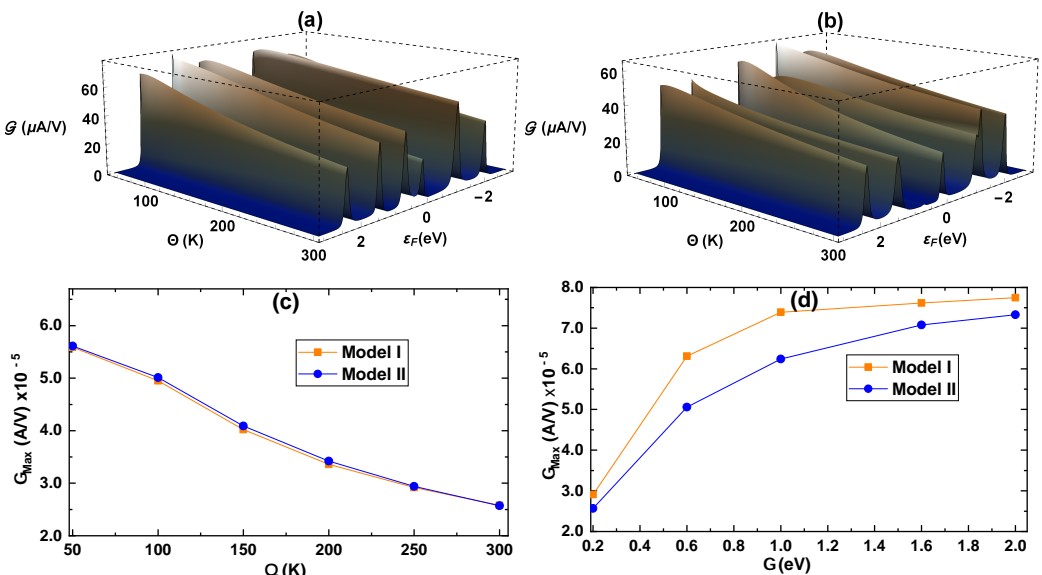

**Figure 8.** (**a**) Electrical conductance (Model I) and (**b**) electrical conductance (Model II) as a function of the temperature ($\Theta$) and Fermi energy ($\varepsilon_F$). (**c**) Maximum electrical conductance as a function of temperature ($\Theta$) and (**d**) maximum electrical conductance as a function of coupling ($\Gamma$) around the Fermi energy ($-0.5$ eV $< \varepsilon_F < 0.5$ eV) for Models I (orange line) and II (blue line).

These maximum values of $\mathcal{G}$ are within the same range for both models, and they steeply decrease as the temperature increases (Figure 8c). This behavior is expected due to power dissipation effects. Notably, the order of magnitude for the calculated electrical conductance agrees with that reported by Kolivoska et al. [16], who studied electrical transport through molecular rods functionalized with catechol and proposed that the low conductance is attributed to molecules trapped in an energy gap.

In contrast, the maximum values of $\mathcal{G}$ increase as the coupling increases ($\Gamma$) due to the hybridization of the discrete states of the molecule with the delocalized states of the contacts. The difference between the two models is an average value of $1 \times 10^{-5}$ A/V over all $\Gamma$ values, being higher for Model I (Figure 8d).

We evaluated the thermal conductance ($\kappa$) at several temperatures ($\Theta$) and various Fermi energy values ($\varepsilon_F$) to understand its dependence on these two factors (Figure 9a,b). In accordance with expectations, we observe that $\kappa$ increases with the increase in temperature, reaching its maximum at 300 K with 246.2 pW/K and 275.9 pW/K for Models I and II, respectively, at $\varepsilon_F = -1.17$ eV. Extracting the maximum values of $\kappa$ with respect to the temperature (Figure 9c) and to the coupling potential ($\Gamma$) at a constant temperature (Figure 9d) within the range of Fermi energies ($-0.5$ eV $< \varepsilon_F < 0.5$ eV), we clearly defined its dependence on these two factors. Looking at Figure 9c, Model I presents an evident proportionality between $\kappa$ and the temperature, whereas, for Model II, $\kappa$ is indirectly proportional to the temperature. This behavior is because the thermal conductance around the Fermi energy $\varepsilon_F = 0$ eV for Model II is very small, and as the energy window around this value increases, $\kappa$ decreases slightly and is almost constant when the temperature rises; then, within this energy range, the temperature has a minor influence on Model II's thermal conductance. When analyzing the dependence of the maximum conductance with respect to $\Gamma$, we observe similar behavior for both models, increasing with higher potentials, resembling what we observed in Figure 8d, where, in this case, $\kappa_{Max} = 560$ pW/K is reached by Model I (see Figure 9d).

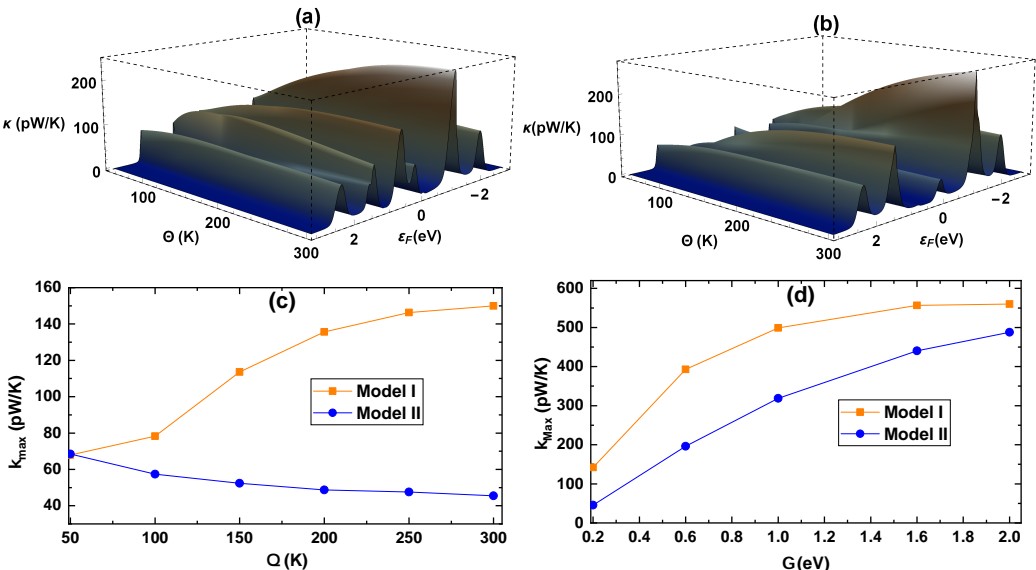

**Figure 9.** (**a**) Thermal conductance (Model I) and (**b**) thermal conductance (Model II) as a function of the temperature ($\Theta$) and Fermi energy ($\varepsilon_F$). (**c**) Maximum thermal conductance as a function of temperature ($\Theta$) and (**d**) maximum thermal conductance as a function of coupling ($\Gamma$) around the Fermi energy ($-0.5$ eV $< \varepsilon_F < 0.5$ eV) for Models I (orange line) and II (blue line).

Continuing with our analysis of the thermoelectrical properties for Models I and II, we determined the Seebeck coefficient ($S$) as a function of Fermi energy ($\varepsilon_F$) and temperature ($\Theta$) (Figure 10a,b). As expected, as the temperature rises, S increases in both models, which is seen more explicitly in Figure 10c, where we depict the maximum of the Seebeck coefficient as a function of the temperature with $\Gamma = 0.2$ eV in an energy range of $-0.5$ eV $< \varepsilon_F < 0.5$ eV. The increment in the Seebeck coefficient is more significant in Model II, reaching its maximum at 391.5 µV/K versus 112.8 µV/K for Model I. This result is because as the temperature increases around the Fermi level $\varepsilon_F = 0$ eV, the transmission probability becomes more asymmetric for Model II, as we previously described. Additionally, when we evaluate the variation in $S_{Max}$ with varying $\Gamma$ coupling (Figure 10d), we find that it steeply decreases for Model II, while it slightly decreases for Model I. This result is directly related to the higher symmetry of the transmission probability around the Fermi energy in stronger coupling regimes.

Finally, we determined the figure of merit (ZT) as a function of temperature ($\Theta$) and Fermi energy (Figure 11a,b). Additionally, for a more explicit analysis, we extracted their maximum values as a function of $\Theta$ (Figure 11c) and $\Gamma$ (Figure 11d). At first sight, we observe a similar behavior to that obtained for S, as ZT is proportional to it. What is interesting to note is that the $ZT_{Max}$ values for both models have the highest performance in the low-coupling regime ($\Gamma = 0.2$ eV), 10 and 14.7 for Model I and Model II, respectively. Conversely, in higher-coupling regimes, $ZT_{Max}$ decreases to a value of about 1.5 for both models. This result is strongly related to the transmission probability values in strong-coupling regimes, where it becomes more symmetrical, and electron transport is more coherent. Therefore, we conclude that the maximum efficiency for the models evaluated, in terms of energy conversion, occurs for the system at the highest evaluated temperature (i.e., $\Theta = 300$ K) and in a weak-coupling regime ($\Gamma = 0.2$ eV).

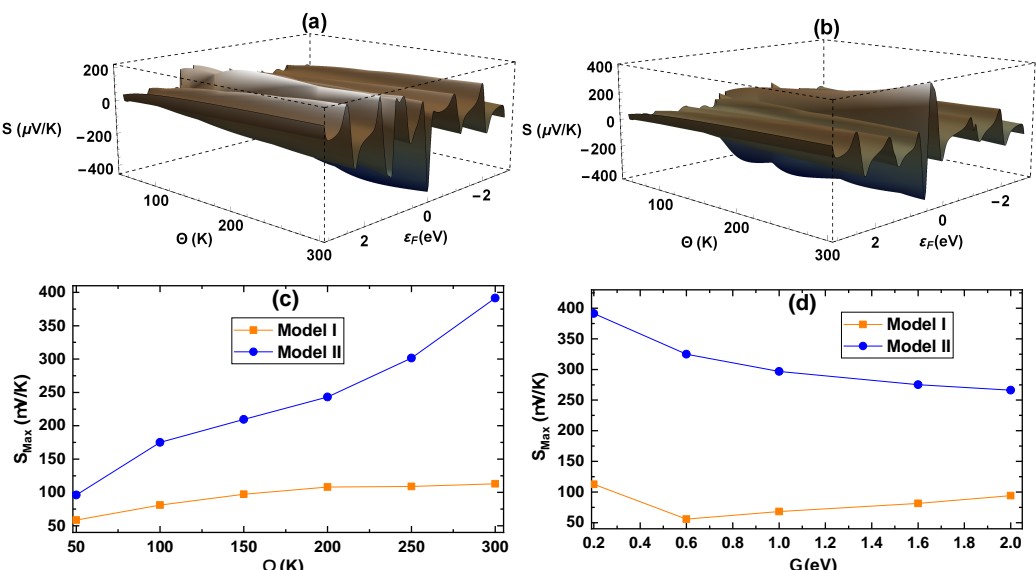

**Figure 10.** (**a**) Seebeck coefficient (Model I) and (**b**) Seebeck coefficient (Model II) as a function of the temperature ($\Theta$) and Fermi energy ($\varepsilon_F$). (**c**) Maximum Seebeck coefficient as a function of temperature ($\Theta$) and (**d**) maximum Seebeck coefficient as a function of coupling ($\Gamma$) around the Fermi energy ($-0.5$ eV $< \varepsilon_F < 0.5$ eV) for Models I (orange line) and II (blue line).

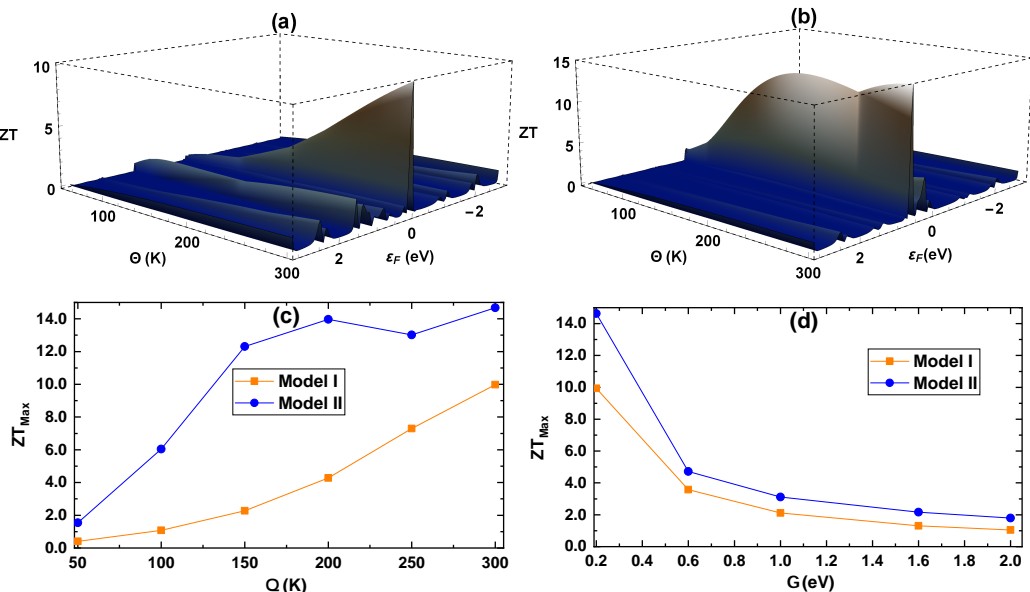

**Figure 11.** (**a**) Figure of merit ($ZT$) (Model I) and (**b**) figure of merit ($ZT$) (Model II) as a function of the temperature ($\Theta$) and Fermi energy ($\varepsilon_F$). (**c**) Maximum figure of merit ($ZT$) as a function of temperature ($\Theta$) and (**d**) maximum figure of merit ($ZT$) as a function of coupling ($\Gamma$) around the Fermi energy ($-0.5$ eV $< \varepsilon_F < 0.5$ eV) for Models I (orange line) and II (blue line).

## 5. Conclusions

We have theoretically determined the thermal and electrical properties of catechol linked to two metal contacts in two models varying in the manner in which they are anchored, herein defined as Models I and II. In order to mathematically define each configuration, we effectively used the Green's function method in the framework of a real-space renormalization process to a linear chain of effective atomic sites.

The results show remarkable differences in the conducting behavior of catechol when comparing it with the single aromatic compound benzene as a reference. After the analysis of the transmission probability, Model I is found to be the system with higher electrical

and thermal conductance. As expected for molecular conducting systems, the electrical conductance is observed to decrease at higher temperatures, whereas it increases when the coupling potential between catechol and the electrodes rises. These results suggest a potential application in diode-type devices since its functionality can be tuned with the nature of this coupling (i.e., $\Gamma$). On the other hand, the thermal conductance differs between the two models: higher conductance is obtained for Model I at higher temperatures, whereas Model II slightly reduces its thermal conductivity when the temperature increases.

Finally, the variations observed in the Seebeck coefficient with the temperature $\Theta$ and the parameter $\Gamma$ can be very useful when it comes to applications in thermoelectric devices or electric power generation, since such variations show, at the same time, the excellent behavior of the figure of merit (ZT), generating a maximum value at the highest evaluated temperature (i.e., 300 K) in Model II in a weak-coupling regime, under conditions suitable for higher energy conversion efficiency for thermoelectric processes in such a device.

We expect that our results will open a new stage for future theoretical and experimental work on improvements in molecular electronics using the conducting properties of aromatic molecules.

**Author Contributions:** E.Y.S.-G. and J.A.G.-C.: Conceptualization, methodology, software, formal analysis, investigation, writing; J.H.O.S. and D.G.: Formal analysis, investigation, supervision, writing; A.L.M., C.A.D. and M.F.H.M.: Formal analysis. All authors have read and agreed to the published version of the manuscript.

**Funding:** J.H.O.S. acknowledges the financial support from Universidad Pedagógica y Tecnológica de Colombia. The authors are grateful to the following Colombian Agencies: CODI-Universidad de Antioquia (Estrategia de Sostenibilidad de la Universidad de Antioquia and projects "Propiedades magneto-ópticas y óptica no lineal en superredes de Grafeno", "Estudio de propiedades ópticas en sistemas semiconductores de dimensiones nanoscópicas", "Propiedades de transporte, espintrónicas y térmicas en el sistema molecular ZincPorfirina", and "Complejos excitónicos y propiedades de transporte en sistemas nanométricos de semiconductores con simetría axial"), and Facultad de Ciencias Exactas y Naturales-Universidad de Antioquia (A.L.M. and C.A.D. exclusive dedication projects 2022–2023). E.Y.S.-G. acknowledges the financial support from "Formación de capital humano de alto nivel Universidad Pedagógica y Tecnológica de Colombia (uptc) nacional, identificado con código bpin 2019000100041, en el marco de la convocatoria n°1 del plan bienal de minciencias del programa de becas de excelencia doctoral del bicentenario. No. 01-2020".

**Data Availability Statement:** No new data were created nor analyzed in this study. Data sharing is not applicable to this article.

**Acknowledgments:** J.H.O.S. Acknowledges to the Centro de Gestión de Investigación y Extensión de la Facultad de Ciencias CIEC-UPTC-Tunja.

**Conflicts of Interest:** The authors declare no conflict of interest.

## Appendix A

*Appendix A.1. Decimation Process for Models I and II*

The reduced geometric configuration that was used in the decimation process for both models is presented in the Figure A1. We label the sites vertically as 1 and 2 and horizontally as *a*, *b*, and *c* to reduce the two-dimensional system to a one-dimensional molecular system. The system was normalized to a line chain of two effective sites, taking all the information toward row a, as shown in Figure 2c (Model I) and Figure 3c (Model II). Later, these effective sites were reduced to a single one, which contains all the information of the original system.

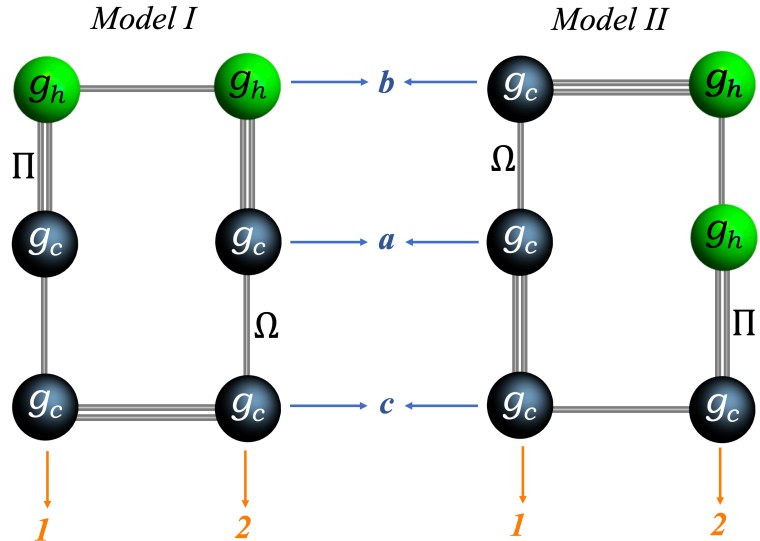

**Figure A1.** Reduced geometric configuration for Models I and II.

The decimation process was carried out through successive applications of the Dyson equation given by Equation (9), which can be written specifically for a system without electrodes as

$$G_{ij} = g_i + g_i V_{i,i-1} G_{j,i-1} + g_i V_{i,i+1} G_{j,i+1}, \tag{A1}$$

where $g_i$ represents the local Green's functions for each atomic site, given by $g_i = 1/(\varepsilon - E_i)$; therefore, $i$ represents the atomic sites for C and OH, while $V_{i,i+1}$ represents the coupling energy between first-neighbor atoms, which, for this molecular system, are labeled as $\Omega$, $\Pi$, and $\Lambda$, respectively.

Thus, in agreement with the atomic sites shown in the reduced geometric configuration in Figures 2b, 3b and A1, the local Green's functions for each atomic site of C and OH are

$$g_C = \frac{1}{(\varepsilon - E_C)}, \tag{A2}$$

and

$$g_{OH} = \frac{1}{(\varepsilon - E_{OH})}, \tag{A3}$$

when the local Green's functions depend of the atomic site energies $E_C$ and $E_{OH}$ and the electron energy $\varepsilon$ when it enters into the molecular system.

The first decimation process was carried out using Equation (A1) to reduce the atomic sites C and OH to a single effective site, labeled as h; thus, we obtain:

$$G_{11} = g_C + g_C \Lambda G_{12}, \tag{A4}$$

and

$$G_{12} = g_{OH} \Lambda G_{11}, \tag{A5}$$

where $\Lambda$ is the coupling energy between the atomic sites C and OH.

Substituting Equation (A5) into Equation (A4), we have

$$G_{11} = \frac{g_C}{(1 - g_C g_{OH} \Lambda^2)} = g_h, \tag{A6}$$

where $g_h$ represents the effective Green's functions when the information from the atomic site OH is carried to the atomic site C. The reduced geometric configurations of these atomic sites for the two models in this study are shown in the Figure A1.

Similarly, to reduce the geometric configuration shown in Figure A1 to a line chain of two effective dots, carrying all the information toward row a, we employ Equation (A1) for Model I, considering the following equations to bring all the information of the system to dot $G_{11}^a$:

$$
\begin{aligned}
G_{11}^a &= g_c + g_c \Pi G_{11}^b + g_c \Omega G_{11}^c \\
G_{11}^b &= g_h \Pi G_{11}^a + g_h \Omega G_{12}^b \\
G_{11}^c &= g_c \Omega G_{11}^a + g_c \Pi G_{12}^c \\
G_{12}^b &= g_h \Pi G_{12}^a + g_h \Omega G_{11}^b \\
G_{12}^c &= g_c \Omega G_{12}^a + g_c \Pi G_{11}^c
\end{aligned}
\tag{A7}
$$

Solving the system of linearly independent equations given in Equation (A7) leads to the effective Green's function $G_{11}^a = \tilde{g}_{1I}$, given by

$$
\tilde{g}_{1I} = \frac{g_c(-1 + g_h^2 \Omega^2)(-1 + g_c^2 \Pi^2)}{-g_h^2 \Omega^2 + \alpha + g_h^3 g_h \Pi^4 + g_c^2(-1 + g_h^2 \Omega^2)(\Omega^2 + \Pi^2)},
\tag{A8}
$$

where $\alpha = 1 - g_c g_h \Pi^2$, and $\tilde{V}_{12}$ is

$$
\tilde{V}_{12} = \frac{\Pi \Omega [g_h^2 \Pi + g_c^2(\Omega - g_h^2 \Omega^3 - g_h \Pi^3)]}{(-1 + g_h^2 \Omega^2)(-1 + g_c^2 \Pi^2)}.
\tag{A9}
$$

In a similar way, we find $\tilde{g}_{2I}$ and $\tilde{V}_{21}$, bringing all the system information to dot $G_{22}^a$. The system of linearly independent equations for this case is

$$
\begin{aligned}
G_{22}^a &= g_c + g_c \Pi G_{22}^b + g_c \Omega G_{22}^c \\
G_{22}^b &= g_h \Pi G_{22}^a + g_h \Omega G_{21}^b \\
G_{22}^c &= g_c \Omega G_{22}^a + g_c \Pi G_{21}^c \\
G_{21}^b &= g_h \Pi G_{21}^a + g_h \Omega G_{22}^b \\
G_{21}^c &= g_c \Omega G_{21}^a + g_c \Pi G_{22}^c
\end{aligned}
\tag{A10}
$$

In this case, the effective Green's functions generated from the combination of equations in (A10) are symmetric (same), where $\tilde{g}_{1I} = \tilde{g}_{2I} = \tilde{g}$ and also $\tilde{V}_{12} = \tilde{V}_{21}$.

The decimation process for Model II was carried out in the same way as Model I. Therefore, $\tilde{g}_1$ and $\tilde{V}_{12}$ were determined at an effective dot $G_{11}^a$ through

$$
\begin{aligned}
G_{11}^a &= g_c + g_c \Omega G_{11}^b + g_c \Pi G_{11}^c \\
G_{11}^b &= g_c \Pi G_{11}^a + g_c \Omega G_{12}^b \\
G_{11}^c &= g_c \Omega G_{11}^a + g_c \Pi G_{12}^c \\
G_{12}^b &= g_h \Omega G_{12}^a + g_h \Omega G_{11}^b \\
G_{12}^c &= g_c \Omega G_{12}^a + g_c \Pi G_{11}^c
\end{aligned}
\tag{A11}
$$

where the effective Green's function $G_{11}^a = g_1$ is given by

$$
\tilde{g}_1 = \frac{g_c(-1 + g_c g_h \Pi^2)(-1 + g_c^2 \Omega^2)}{\alpha + g_c^4 \Omega^4 + g_c^3 g_h \Pi^2(\Pi^2 + \Omega^2) - g_c^2(\Pi^2 + 2\Omega^2)},
\tag{A12}
$$

and the effective coupling $\tilde{V}_{12}$ is

$$
\tilde{V}_{12} = \frac{g_c \Pi \Omega (g_c \Pi + g_h \Omega - g_c^2 g_h(\Pi^3 + \Omega^3))}{(-1 + g_c g_h \Pi^2)(-1 + g_c^2 \Omega^2)}.
\tag{A13}
$$

Then, in the same way, $\tilde{g}_2$ and $\tilde{V}_{12}$ are carried at an effective dot $G_{22}^a$:

$$
\begin{aligned}
G_{22}^a &= g_h + g_h \Omega G_{22}^b + g_h \Pi G_{22}^c \\
G_{22}^b &= g_h \Omega G_{22}^a + g_h \Pi G_{21}^b \\
G_{22}^c &= g_c \Pi G_{22}^a + g_c \Omega G_{21}^c \\
G_{21}^b &= g_c \Omega G_{21}^a + g_c \Pi G_{22}^b \\
G_{21}^c &= g_c \Pi G_{21}^a + g_c \Omega G_{22}^c
\end{aligned}
\tag{A14}
$$

Developing the system of equations given by Equation (A14), $\tilde{g}_2 = G_{22}^a$ is obtained as

$$
\tilde{g}_2 = \frac{g_h(-1 + g_c g_h \Pi^2)(-1 + g_c^2 \Omega^2)}{1 - 2g_c g_h \Pi^2 - g_h^2 \Omega^2 + g_c^2 g_h \Pi^2 \Omega^2 + g_c^2(-\Omega^2 + g_h^2 \gamma)} \, ,
\tag{A15}
$$

where $\gamma = (\Pi^4 + \Omega^4)$.

Now, applying Equation (A1) in the effective line chain for the first and second effective dots of both models, we obtain:

$$
G_{11}^a = \tilde{g} + \tilde{g}\tilde{V}_{12}G_{12}^a \, ,
\tag{A16}
$$

and

$$
G_{22}^a = \tilde{g} + \tilde{g}\tilde{V}_{21}G_{12}^a \, .
\tag{A17}
$$

Substituting Equations (A16) and (A17), we have

$$
G_{11}^a = \frac{\tilde{g}}{(1 - \tilde{g}^2 \tilde{V}_{12}\tilde{V}_{21})} \, ,
\tag{A18}
$$

and

$$
G_{12}^a = \frac{\tilde{g}^2 \tilde{V}_{12}}{(1 - \tilde{g}^2 \tilde{V}_{12}\tilde{V}_{21})} \, .
\tag{A19}
$$

In a similar way, for Model II, we obtain

$$
G_{11}^a = \tilde{g}_1 + \tilde{g}_1 \tilde{V}_{12}G_{12}^a \, ,
\tag{A20}
$$

and

$$
G_{22}^a = \tilde{g}_2 + \tilde{g}_2 \tilde{V}_{21}G_{12}^a \, .
\tag{A21}
$$

Substituting Equations (A20) and (A21), we have

$$
G_{11}^a = \frac{\tilde{g}_1}{(1 - \tilde{g}_1 \tilde{g}_2 \tilde{V}_{12}\tilde{V}_{21})} \, ,
\tag{A22}
$$

and

$$
G_{12}^a = \frac{\tilde{g}_1 \tilde{g}_2 \tilde{V}_{12}}{(1 - \tilde{g}_1 \tilde{g}_2 \tilde{V}_{12}\tilde{V}_{21})} \, .
\tag{A23}
$$

*Appendix A.2. Eigenvalues*

The system of eigenvalues can be calculated by means of the Hamiltonian of the contact-independent molecular system. Considering an interaction only with the first neighbors, this Hamiltonian can be written cleanly as:

$$
H_{cat} = \sum_i t_{i,i+1}|i\rangle\langle i+1| + t_{i+1,i}|i+1\rangle\langle i| + \sum_i E_i|i\rangle\langle i| \, ,
\tag{A24}
$$

As previously mentioned, each $t_j$ represents the coupling between the first neighboring atoms in the system. For this model, we have taken $\Omega$ as a simple bond ($\sigma$-bond) between C-C atoms, $\Pi$ as a double bond ($\pi$-bond) between C=C atoms, and $\Lambda$ as a $\sigma$-bond in C-OH. Also, $E_i$ is the energy of the atomic sites, denoted in this case as $E_C$ and $E_{OH}$ for the C and

OH atomic sites, respectively. In Equation (A24), each function $|i\rangle$ can take eight values, corresponding to the atomic positions of the six carbon atoms and the two OH systems, as shown in Figure 1. At this point, it is possible to easily calculate each of the matrix elements of the Hamiltonian given in Equation (A24) and thus obtain its matrix representation as an $8 \times 8$ matrix,

$$H_{cat} = \begin{pmatrix} E_c & \Omega & 0 & 0 & 0 & 0 & 0 & \Pi \\ \Omega & E_c & 0 & \Pi & 0 & 0 & 0 & 0 \\ 0 & 0 & E_{oh} & \Lambda & 0 & 0 & 0 & 0 \\ 0 & \Pi & \Lambda & E_c & \Omega & 0 & 0 & 0 \\ 0 & 0 & 0 & \Omega & E_c & \Lambda & \Pi & 0 \\ 0 & 0 & 0 & 0 & \Lambda & E_{oh} & 0 & 0 \\ 0 & 0 & 0 & 0 & \Pi & 0 & E_c & 0 \\ \Pi & 0 & 0 & 0 & 0 & 0 & \Omega & E_c \end{pmatrix} \tag{A25}$$

This matrix can be diagonalized by any elementary method of linear algebra to obtain the set of eigenvalues corresponding to the catechol molecular system. As already mentioned, in the development of this work, the values obtained through this procedure were 2.26 eV, $-2.15$ eV, 1.37 eV, 1.33 eV, $-1.24$ eV, $-1.13$ eV, 0.57 eV, and 0.0 eV.

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
