# Peer review of "Theoretical Study of Electronic and Thermal Transport Properties through a Single-Molecule Junction of Catechol"

_condensedmatter, doi:10.3390/condmat8030060_

Round 1

Reviewer 1 Report

Please, see the attached file above

Author Response

Reviewer 1:

Comments:

In this work, the authors have studied electrical and thermal transport properties of catechol using Green’s functions renormalized in a real space domain within a framework of tightbinding approximation to first neighbors. The manuscript is well written and the methodology is detailed enough to be reproducible. The authors found significant differences between the transport properties of benzene and catechol but also non negligible differences depending of how the catechol anchored. I think the manuscript is interesting but it need some revision.

Our answer

We want to thank and express our gratitude to the Reviewer for his/her excellent report, which we believe has helped us substantially improve the quality and clarity of our manuscript.

Issue 1.1. Presentation. Figures are all well designed and most of text is correct but I could find some typos. For instance, Figure 4 and 9 captions.

Our answer . Typos have been corrected throughout the manuscript.

Issue 1.2. Discussion. The manuscript is mostly descriptive. The authors should discuss in detail the physical-chemical origin of their results. What is the origin of the differences they found between models?

Our answer. Oddly, the reviewer made this point since we discuss the physical-chemical origin of the results and differences between both models in two paragraphs of the manuscript (Page 8, lines 205-223, 1st submission):

“The resonant peak of T(ε) around the Fermi level (ε = 0) for both models represents the allowed electronic states at that level, thus, clearing the path for the transport of charge carriers due to an increase in the density of states. Therefore, the forbidden band between the highest occupied molecular orbital (HOMO) and the lowest unoccupied molecular orbital (LUMO) tends to cancel out, pointing out that under these conditions, catechol acts as a conducting material. This result contrasts the simpler aromatic system, such as a benzene molecule, which behaves as an semiconducting material (Figure 4, brown dashed line) with a gap around the Fermi level εF=0 eV [24 ,25 ]. Since the OH groups alter the hybridization on the aromaticity of the molecular system, this clearly enhances the density of states around the Fermi level, turning it from a semiconducting to a conducting molecular skeleton.”

“While comparing the distinct behavior of both models, we can find evidence that the anchored fashion of the catechol’s OH groups to the leads alters the resistance of the conducting molecular system. Thus, the resonant peaks in Model I are broader than for Model II, although their eingenvalues are very close. This difference is strongly related to that in Model I, none of the OH groups are attached to the leads, indicating more diffusion probability of the charge carriers along the molecular system, whereas with one OH group attached to a lead (Model II), we can observe a lower area under the curve, indicating a more significant resistance to the conductivity. “

Furthermore, we have added a paragraph, where the physical origin of the difference between both models is observed, highlighting the dependence of the electrical and thermal properties with the electronic transmission probability. The paragraph included in the manuscript is on page 12 in lines 274-282:

Thus, we can observe in Figure 7 a marked difference between models I and II at the value of εF = 0.5 eV in the electrical conductance (?), the thermal conductance (κ), Seebeck coefficient (S), and figure of merit (ZT). This result strongly agrees with the transmission probability (Figure 4) since its value evaluated at ε = 0.5 eV presents an antiresonance for model II, where transmission probability decreases considerably and reaches almost zero, whereas model I has a non-zero transmission. In this sense, according to equations (18-21), the electrical and thermal properties depend on the transmission integral (ζn) evaluated in the Fermi energy, therefore at εF = 0.5 eV, the highest difference in both models is shown in each of these properties (Figure 7).

Issue 1.3. Vibrations. If I understood properly the authors do not calculate the thermal transport due to vibrations. They should make at least a comment on that.

Our answer. We thank the reviewer to point out this issue. To clarify it we added the following sentence in the manuscript (Page 7, lines 188-192, Resumitted version): 

Summing up, equations (18-21) show the dependence of the electrical and thermal properties on the probability of transmission. Noteworthy, the energy transport in the form of heat (κ) depends on the sum of the electrical (κel) and phononic (κph) contributions. However, for the molecular systems studied in this work, κel is much larger than κph, therefore, the contributions of κph to thermo-electric transport are negligible. [35,37]

Reviewer 2 Report

The authors present an article where  they have theoretically determined the thermal and electrical properties of catechol linked to two metal contacts using two models. The results could very usefull for creating novel thermoelectric devices

There are many typos in the text like in the Fig.4 caption ''Fig. 4 ransmission"

Author Response

Reviewer 2:

Comments:

The authors present an article where they have theoretically determined the thermal and electrical properties of catechol linked to two metal contacts using two models. The results could very usefull for creating novel thermoelectric devices.

Our answer

We want to thank and express our gratitude to the Reviewer for his/her excellent report, which we believe has helped us substantially improve the quality and clarity of our manuscript.

Issue 1.1. There are many typos in the text like in the Fig.4 caption ''Fig. 4 ransmission"

Our answer. Typos have been corrected throughout the manuscript.

Round 2

Reviewer 1 Report

The authors have addressed all my comments and concerns.